# Biostimulating Gut Microbiome with Bilberry Anthocyanin Combo to Enhance Anti-PD-L1 Efficiency against Murine Colon Cancer

**DOI:** 10.3390/microorganisms8020175

**Published:** 2020-01-25

**Authors:** Xuerun Liu, Luoyang Wang, Nan Jing, Guoqiang Jiang, Zheng Liu

**Affiliations:** Key Lab of Industrial Biocatalysis, Ministry of Education, Department of Chemical Engineering, Tsinghua University, Beijing 100084, China; liu-xr17@mails.tsinghua.edu.cn (X.L.); jn17@mails.tsinghua.edu.cn (N.J.); jianggq@mail.tsinghua.edu.cn (G.J.)

**Keywords:** immune checkpoint inhibitors, immunotherapy, gut microbiome, bilberry anthocyanins, low molecular citrus pectin, chitosan

## Abstract

Recent advances have revealed the essential role of gut microbiomes in the therapeutic efficiency of immune checkpoint inhibitors (ICIs). Inspired by biostimulation, a method using nutrients to accelerate the growth of soil microorganisms and the recovery of soil microbial consortia, here we propose a bilberry anthocyanin combo containing chitosan and low molecular citrus pectin (LCP), in which LCP–chitosan is used to encapsulate anthocyanins so to enhance its digestive stability and, moreover, modulate the microbiome more favorable for the PD-L1 blockade treatment. Using murine MC38 colon cancer as a model system, we examined the effects of the combo on modulating the gut microbiome and therapeutic efficiency of PD-L1 blockade treatment. It was shown that bilberry anthocyanins enriched the subdominant species, increased both the concentration and the proportion of butyrate in feces and enhanced intratumoral CD8^+^ T cell infiltrations. The application of the bilberry anthocyanin combo restored the species diversity of gut microbiome decreased by LCP–chitosan and achieved the best control of tumor growth. These preliminary results indicated unprecedented opportunities of probiotics combo in improving the therapeutic efficiency of immune checkpoint inhibitor through manipulating gut microbiome.

## 1. Introduction

While immunotherapy through targeting immune checkpoints to unleash the adaptive immune response has rekindled hopes for millions of cancer patients, the response rates are below 30% for immune checkpoint inhibitors (ICIs), such as anti-programmed death 1 (PD-1) and anti-programmed death ligand 1 (PD-L1), in some tumor types [1,2,3,4]. More observations of the differences in gut microbiomes between responders and nonresponders, and, moreover, positive results in recapitulating the patient phenotype [5,6,7] through fecal microbiota transplant (FMT) have indicated the potential of manipulating the gut microbiome to enhance therapeutic response [7,8,9]. Although the underlying mechanisms remain to be discovered and differentially enriched bacterial taxa in responder patients are observed across cohorts, the significance of the gut microbiome for the therapeutic outcomes of ICIs has been recognized [10,11,12,13].

Anthocyanins are polyphenols with proven health attributes, for example, retinal protection [14]. Recently, the prebiotic function of anthocyanins has also attracted growing attention. Both in vitro fermentation of anthocyanins by human fecal microbiome [15] and in vivo dietary intervention in mice [16,17] have demonstrated that anthocyanins can improve the diversity of the gut microbiome, promote growth of the beneficial bacterial groups while inhibit the potentially harmful bacterial groups, and increase the concentration of fecal short-chain fatty acids (SCFAs) [16,18]. All these indicate a high potential of anthocyanins in enhancing therapeutic efficiency of α-PD-L1. However, the observation of the low recovery (below 27%) in ileal effluent [19,20] suggests the poor availability of anthocyanins in the colon and thus the need to enhance the digestive stability of anthocyanin products used for oral administration.

Biostimulation is a kind of soil remediation technique that utilizes nutrients to accelerate the growth of indigenous microorganisms to degrade contaminates [21]. An efficiency index to soil remediation is the recovery of microbial consortia that underpins the cultivation capability of soil [22]. We propose that the concept of biostimulation is also applicable for manipulating gut microbiome that is favorable for therapeutic treatment using ICIs. With appreciation of the proven effectiveness of anthocyanins in modulating the gut microbiome favorable for ICIs treatment—yet recognizing its poor digestive stability—we propose an anthocyanin combo for biostimulating the gut microbiome in order to enhance the therapeutic efficiency of the ICIs. Chitosan and low molecular citrus pectin (LCP) were added to the combo, which formed complex encapsulating anthocyanins to inhibit gastrointestinal digestion and, consequently, enhance its performance in manipulating the gut microbiome. Moreover, we consider LCP, a kind of diary fiber, as one prebiotic candidate [23,24] that might also be engaged in manipulating gut microbiomes. In the present study, we examined the stimulating effects of bilberry anthocyanins and bilberry anthocyanin combo on the antitumor efficiency of the PD-L1 blockade in the mouse MC38 colon tumor model, respectively. We monitored the mice gut microbiome, fecal short-chain fatty acids and tumor T lymphocyte infiltration in the above-mentioned treatments to explore the underlying mechanisms of the anthocyanin combo-enhanced therapeutic efficiency of PD-L1 blockade treatment.

## 2. Materials and Methods

### 2.1. Chemicals and Antibodies

Standardized bilberry extract (MIRTOSELECT^®^) was purchased from Indena S.p.A. (Milan, Italy) containing 36% of anthocyanins. The composition of the anthocyanins was determined by high-performance liquid chromatography–ultraviolet-visible spectroscopy-tandem mass spectrometry method (HPLC-UV/vis-MS/MS), as detailed below. Low molecular citrus pectin (LCP) with the molecular weight range of 5000 to 15000 Daltons was purchased from Zhejiang Gold Kropn Bio-technology Co., Ltd. (Zhejiang, China). Chitosan (C105799) was purchased from Aladdin Co. Ltd. (Shanghai, China). α-PD-L1 monoclonal antibody (mAb) (10F.9G2) and immunoglobulin G2b (IgG2b, BE0090) isotype were purchased from BioXCell (West Lebanon, NH, USA).

### 2.2. Analysis of Anthocyanins by HPLC-UV/vis-MS/MS

HPLC-UV/vis-MS/MS analysis of anthocyanins was performed using a Shimadzu LCMS-8040 liquid chromatography with mass spectrometry (Shimadzu, Kyoto, Japan), using a TSKgel ODS-100V column (5 μm, 250 × 4.6 mm, Tosoh Corporation, Tokyo, Japan) maintained at 35 °C. During a run, 10 μL bilberry extracts of 5 mg/mL was injected. A gradient mode of elution was performed with 1.0% (v/v) formic acid (phase A) and methanol (phase B) at a flow rate of 1 mL/min. The gradient program was set as the following: 7%–10% B from 0 to 2 min, 10%–34% B from 2 to 22.5 min, isocratic 34% B for 2.5 min, 34%–70% B from 25 to 55 min, 70%–7% B from 55 to 60 min, and held in isocratic 7% B for 5 min to clean the column. Eluents were detected at 520 nm (anthocyanin absorption maximum). Nitrogen was used as the nebulizing and drying gas, 3.0 L/min and 15.0 L/min respectively; ion spray voltage was set at 4.5 kV. Electron spray ionization mass spectra (ESI-MS) were scanned from m/z 100 to 600. Spectra were acquired in the positive ion mode for anthocyanins. Anthocyanins were identified according to their retention times and elution order in HPLC-UV/vis, followed by a comparison of their mass spectra in HPLC-UV/vis-MS/MS against the data in the literature [25]. Cyanidin 3-O-glucoside was further confirmed by the retention time of the available authentic standard.

### 2.3. In Vitro Digestion Procedure

Gastrointestinal digestion was conducted as previously described [26]. Firstly, we prepared 20 mL aqueous solution containing bilberry extracts (1mg/mL), chitosan (2 mg/mL) and LCP (4 mg/mL), and the pH was adjusted to 2.0 with HCl. Then 3144 units of pepsin were added and the sample was incubated in a 37 °C water bath for 2 h. Aliquots of 10 mL were collected and centrifuged using a Thermo Fisher Scientific Sorvall LYNX 4000 Centrifuge (Thermo Fisher Scientific, Waltham, MA, USA) at 26,000× *g* at 4 °C for 10 min. The supernatant was collected for determining total anthocyanin content. The anthocyanin–LCP–chitosan complexes were also collected and washed using an HCl solution of pH 2.0 three times, then the total anthocyanin content in the washing solution was determined. The pH of the remaining 10 mL from the gastric digestion step was adjusted to pH 6.8 with 0.5 M NaHCO_3_. Afterwards, 1.25 mL pancreatin (2 g/L, 8× USP specifications, Sigma-Aldrich, St. Louis, MO, USA) and 1.25 mL of bile salts (25 g/L, Sigma-Aldrich, St. Louis, MO, USA) were added to each vessel and incubated for 4 h. At 2 h and 4 h, the total anthocyanin content in supernatant and anthocyanin-LCP–chitosan complexes were both determined. The total anthocyanin content of bilberry extract before and after digestion was determined by the pH-differential spectrophotometric method [27]. Each experiment was conducted in triplicate.

### 2.4. Preparation of Anthocyanin Combo

Buffer solutions (0.2 M sodium acetate-acetic acid buffer) were prepared at pH 2.5, 3.5, 4.5 and 5.5. All the chemicals including LCP, chitosan, and anthocyanins were dissolved in the buffer solution prior to use. LCP and anthocyanins mixture at a volume ratio of 1:1 were added into the chitosan solution and incubated at 4 °C for 30 min under constant stirring, the final concentration of chitosan, pectin and bilberry extracts was 1, 2 and 0.5 mg/mL, respectively. The electrostatic interaction between negatively charged LCP and positively charged chitosan may form complex encapsulating anthocyanin, thus preventing enzymatic digestion. The residual anthocyanins in the solution were recovered by centrifugation at 8000 rpm at 4 °C for 10 min and determined using the pH-differential spectrophotometric method [27]. The encapsulation efficiency (EE, %) was calculated using the following equation: (1)$$EE\left(\%\right)=\frac{total\;amount\;of\;anthocyanins-free\;anthocyanins}{total\;amount\;of\;anthocyanins}\times 100.$$

### 2.5. Animals and Treatment

Female C57BL/6 mice (eight weeks old) were purchased from the Vital River Laboratory Animal Technology Co. Ltd. (Beijing, China). During the experimental session, all mice were housed under pathogen-free conditions in the animal care facilities at the Institute of Biophysics, Chinese Academy of Sciences. After a one-week adaptation, mice were subcutaneously inoculated with 5 × 10^5^ MC38-OVA cells at the right flank. MC38 was from American Type Culture Collection (ATCC), and was cultured in 5% CO_2_ and maintained in Dulbecco’s Modified Eagle’s Medium (Invitrogen, Carlsbad, CA, USA) supplemented with 10% heat-inactivated fetal bovine serum (HyClone, Logan, UT, USA). MC38 cells were suspended again in phosphate buffered saline (PBS) for subcutaneous injection. Tumor volumes were measured every 3–4 days with an electronic caliper and reported as volume using the formula (width^2^ × length/2). Seven days after tumor inoculation, mice were randomized by tumor size into experimental groups of 6 animals. For α-PD-L1 treated groups, mice received twice intraperitoneal (i.p.) injection of 200 mg α-PD-L1 dissolving in glucose injection at days 7 and 10. For the IgG2b isotype control, mice received isotype IgG2b at the same time point. For the α-PD-L1/LCP–chitosan treatment group, mice were given a dose of 100 mg pectin/kg body weight and 50 mg chitosan/kg body weight every day. For the α-PD-L1/anthocyanin treatment group, mice were given a dose of 25 mg bilberry extracts/kg body weight every day. For the α-PD-L1/anthocyanin combo treatment group, mice were given a dose of 25 mg bilberry extracts/kg body weight, 100 mg pectin/kg body weight and 50 mg chitosan/kg body weight every day. For all groups, gavage started from day 7 following tumor inoculation. All animal procedures were performed in accordance with institutional guidelines and regulations and approved by the Institutional Animal Care and Use Committee of Institute of Biophysics, Chinese Academy of Sciences (approval ID SYXK2018-35).

### 2.6. DNA Extraction and Bacterial Identification in Fecal Samples

After 2 weeks of treatment, the feces of mice were collected and frozen at −80 °C. Bacterial DNA from the feces of mice was extracted using the QIAamp DNA Stool Mini Kit (Qiagen, Hilden, Germany) according to the manufacturer’s instructions. The V3–V4 hypervariable region of 16S rRNA gene was amplified with universal primers: 338F (5′-ACTCCTACGGGAGGCAGCA-3′) and 806R (5′-GGACTACHVGGGTWTCTAAT-3′) by PCR (ABI GeneAmp 9700, Applied Biosystems, Foster City, CA, USA) with a 20 μL reaction system (TransGen, Beijing, China) containing 4 μL 5× Fast Pfu Buffer, 2 μL 2.5mM dNTPs, 0.8 μL Forward Primer (5 μM), 0.8 μL Reverse Primer (5 μM), 0.4 μL Fast Pfu Polymerase, 0.2 μL BSA, and 10 ng template DNA. The procedure was as follows: 3 min of denaturation at 95 °C, 27 cycles of 30 s at 95 °C (denaturation), 30 s for annealing at 55°, 45 s at 72 °C (elongation), and a final extension at 72 °C for 10 min. PCR products were examined by 2% agarose electrophoresis and purified using the AxyPrep DNA Gel Extraction Kit (Axygen Biosciences, Union City, CA, USA). Then, the PCR products were quantified using QuantiFluor-ST (Promega, Madison, WI, USA) and sequenced with an Illumina MiSeq platform according to the standard protocols (Shanghai Majorbio Bio-Pharm Technology Co. Ltd., Shanghai, China). The generated raw reads were screened using FASTP tool with default parameter for quality control and removal of low-quality sequences. The remaining high-quality sequences were clustered into operational taxonomic units (OTUs) at 97% of identity by using USEARCH (Version 8.1, http://www.drive5.com/usearch). Taxonomic classification was conducted using the ribosomal database project (RDP) classifier (Version 2.2, http://www.sourceforge.net/projects/rdp-classifier/) via the Silva database (Release132, http://www.arb-silva.de) with a confidence threshold of 70%. Raw sequence reads of the 16S rRNA gene amplicon data are available through the SRA with accession number PRJNA598985.

### 2.7. Bioinformatics Analysis

The alpha diversity was calculated respectively for each group by package vegan and plotted with package ggplot2 in R (version, R 3.6.1). Median values and interquartile ranges are shown in boxplots. For beta-diversity (sample-to-sample dissimilarity), the principal component analysis (PCA) was performed using R package ade4 and illustrated with R package made4. The key OTUs that affect PCA clustering was extracted from result of PCA analysis and plotted with ggplot2. Enterotype analysis was performed using Calinski–Harabasz index as an indicator of optimal clustering in R with packages cluster, clusterSim and ade4. Linear discriminant analysis (LDA) effect size (LEfSe) were performed to investigate differences in the community composition between each enterotype (http://huttenhower.sph.harvard.edu/galaxy). PICRUSt (Phylogenetic Investigation of Communities by Reconstruction of Unobserved States) was used to evaluate the functional potential of microbial communities [28]. Briefly, BIOM format of data were constructed using QIIME against the Greengenes 13.5 database, then processed with the PICRUSt software using the EggNOG (evolutionary genealogy of genes: Non-supervised Orthologous Groups, http://eggnog.embl.de/) databases to construct the abundance tables of COGs.

### 2.8. Quantification of Fecal Short-Chain Fatty Acids (SCFAs) by Gas Chromatography

Feces samples of 0.1 g were suspended in 0.4 mL water and 0.1 mL 50% sulphuric acid, homogenized for 5 min and centrifuged at 13,000× *g* for 10 min. The supernatant was extracted with ethyl ether of equal volume and the suspension was obtained for gas chromatography on Shimadzu GC-2010 system (Shimadzu, Kyoto, Japan) equipped with a flame ionization detector (FID). Separation was achieved using a HP-INNOWax column (30 m × 0.250 mm × 0.25 μm, Agilent Technologies Inc., Santa Clara, CA, USA). Split ratio was 10:1, the pressure of carrier gas, helium, was maintained at 100 kPa. Injection volume was 1 μL. Helium (30 mL/min), hydrogen (40 mL/min) and dry air (400 mL/min) were used as auxiliary gases for the flame ionization detector. The injector and detector temperatures were 250 °C and 280 °C, respectively. The oven temperature was held at 60 °C at first, then increased to 100 °C at a rate of 20 °C /min, and maintained for 3 min, finally to 210 °C at a rate of 30 °C /min and maintained for 5 min.

### 2.9. Quantification of Immune Cells in Tumor Tissues by Flow Cytometry

Tumors were excised on day 14 after inoculation. The isolated tumor tissues were digested with Collagenase IV (Yeasen Biotech Co., Ltd., Shanghai, China) at 37 °C for 2 h, and then filtered through a 70-μm cell strainer (Corning Incorporated, Corning, NY, USA) to obtain the single cells. The harvested cells were washed twice with PBS, and then stained in the dark with anti-mouse antibodies for CD4 (GK1.5), CD8 (53-6.7), CD45 (30-F11) (BioLegend, San Diego, CA, USA) at 4 °C for 30 min. Analysis was performed immediately by using an FACS Calibur flow cytometer (Becton-Dickinson, Fullerton, CA, USA).

### 2.10. Statistical Analysis

The statistical analysis was performed by GraphPad Prism 8 with two-tailed Student’s *t*-test or repeated measures ANOVA (time × tumor volume) with Sidak′s multiple comparisons test. All data were presented as mean ± SD. *p* < 0.05 were considered to be statistically remarkable, while *p* < 0.01 indicate that the differences reach the levels of statistical high significance.

## 3. Results

### 3.1. The Protection Effects of Chitosan/Pectin for Anthocyanins Against Gastrointestinal Digestion

Parent ions, and daughter ions of detected anthocyanins are summarized in Table 1. Bilberry anthocyanins contain 20 major monomeric anthocyanins, including five anthocyanidins (delphinidin, cyanidin, petunidin, peonidin and malvidin) and their monoglycoside derivatives [29]. When incubated in simulated gastric fluid without pepsin, the total content of both anthocyanins and polyphenols remained constant throughout 2 h (Appendix A), and even kept stable when pH raised up to 5.5 (Appendix A). In the case of simulated intestinal fluid with pancreatin, the total content of anthocyanins and polyphenols decreased to 47.3% and 91.5%, respectively, after 4 h incubation (Appendix A), suggesting the poor stability of anthocyanins in the small intestine. As shown in Figure 1B, chitosan and LCP forms polyelectrolyte complexes at different pHs (2.5, 3.5, 4.5 and 5.5) encapsulating anthocyanins, in which the maximum encapsulation efficiency of 47% is achieved at pH 3.5. It is noted that chitosan is soluble in simulated gastric fluid (SGF) but dissoluble in simulated intestinal fluid (SIF). It is anticipated that chitosan will form precipitate encapsulating anthocyanins when the pH increases from simulated gastric fluid to simulated intestinal fluid. Indeed, our experiment demonstrated that chitosan could enhance the digestive stability of anthocyanins (Appendix A). Similarly, the encapsulation by LCP–chitosan also leads to an improved stability of anthocyanins in simulated gastrointestinal fluid (Figure 1C). All these suggest an effective protection of anthocyanins from the gastrointestinal digestion, which is favorable for the delivery of anthocyanin to the colon.

### 3.2. Effects of Anthocyanin and Anthocyanin Combo on the Anti-PD-L1 Efficiency in Mouse MC38 Colon Tumor

Here we examined the effects of anthocyanin, anthocyanin combo on the α-PD-L1 treatment. It is anticipated that the encapsulation by LCP–chitosan may deliver more anthocyanin to colon and thus offers an intensified enhancement. Meanwhile LCP, which could induce SCFA formation [23] and promote the growth of beneficial gut bacteria [24], may also impact on the gut microbiome and thus the α-PD-L1 treatment. As illustrated in Figure 2A, twice intraperitoneal injection (i.p.) of α-PD-L1 mAb on the 7th and 10th day after tumor inoculation caused tumor growth delay compared to the control group. Oral supplement with anthocyanins could improve the therapeutic effects of α-PD-L1 mAb. Moreover, the application of the anthocyanin combo further increased therapeutic effectivity of α-PD-L1 mAb. In comparison, solely supplement with LCP–chitosan achieved no obvious benefits.

### 3.3. Diversity of the Gut Microbiome of Mice with Different Treatment

Mice feces samples in above study were collected for microbial diversity analysis of 16S rRNA after 2 weeks of treatment. Firstly, we performed analysis of beta diversity at OUT level. As shown in Figure 2B, the principal component analysis (PCA) showed significant changes (Appendix A, Adonis variance analysis based on Bray–Curtis distance matrices at OTU level) in the composition of gut microbiome after oral supplement with LCP–chitosan and anthocyanin combo. There were 5 key OTUs that affect the PCA clustering, i.e., OTU55, OTU63, OTU24, OTU415, OTU515 (Figure 2C). OTU55 belonging to *Muribaculaceae* was significantly enriched in the α-PD-L1/LCP–chitosan group (*p* = 0.04342, Kruskal–Wallis test), while OTU415 affiliated to *Lachnospiraceae* was enriched in the anthocyanin combo group. Further analyses indicate an enrichment of *Muribaculaceae* in the α-PD-L1/LCP–chitosan group (Figure 2D), which is the most abundant commensal at the family level.

Higher species diversity is correlated with higher response rates to ICIs [30,31], hence we analyzed the alpha diversity. As shown in Figure 2E, compared with the control group or the α-PD-L1 group, the alpha diversity indices of the communities from samples of the α-PD-L1/LCP–chitosan group were significantly decreased, indicating a profound impact of LCP–chitosan on mice gut microbiome. Interestingly, the alpha diversity indices of samples from anthocyanin combo group returned to the level of control group, hinting a synergy effects between anthocyanins, LCP and chitosan. Further analysis shows that LCP–chitosan obviously decease the subdominant species diversity, while bilberry anthocyanins enrich the subdominant species and recover LCP–chitosan induced decrease in these species (Figure 2F).

### 3.4. Analysis of Differential Taxons

We then compared the community structure of each group at different taxonomic levels. As shown in Figure 3A,B, at the phylum level, the *Firmicutes* to *Bacteroidetes* ratio (F/B) decreased in the α-PD-L1 group compared to the control group, and further decreased when combining with LCP–chitosan, while increased when combining with bilberry anthocyanins though there are no significant differences between each of them. Besides, the F/B ratio in α-PD-L1/anthocyanin combo group returned to the level of control group, hinting at possible improvement in energy harvest from food for hosts [32]. At the family level, *Muribaculaceae* and *Lachnospiraceae* were the most dominant commensals in the gut microbiome across all groups (Figure 3C). Compared with the α-PD-L1 group, the abundance of *Lachnospiraceae* increased in the α-PD-L1/anthocyanin group and the α-PD-L1/anthocyanin combo group while the abundance of *Muribaculaceae* decreased.

### 3.5. Enterotype Analysis and Functional Differences in the Gut Microbiome

All the samples can be clustered into three groups using the Jensen–Shannon Distance (JSD) at the OTUs level (Figure 4A). Enterotype 1 was mainly from the α-PD-L1/anthocyanin group and the α-PD-L1/anthocyanin combo group, enterotype 2 was from the control group and the α-PD-L1 group, enterotype 3 from the α-PD-L1/LCP–chitosan group. LEfSe (Linear discriminant analysis Effect Size) analysis showed differential gut microbiome signatures between these three enterotypes (Figure 4B). *Ruminococcaceae* and *Lachnospiraceae* were enriched in enterotype1, while *Muribaculaceae* was enriched in enterotype3. Subsequently, we performed the analysis of 16S-predicted functional profiles for studying the information behind the differences of the gut microbiome structure among clusters. As shown in Figure 4C, some gene functions show significant differences, among which carbohydrate transport and metabolism was significantly higher in enterotype1 and enterotype2 compared to enterotype3 (Figure 4D). Since one of the major factors shaping the gut microbiome is the influx of dietary fiber into the intestine [33], and the major metabolites of dietary fiber, such as SCFAs, are important for maintaining host health [34], increasing in carbohydrate transport and metabolism in enterotype1 and enterotype2 hints possible improvement in energy metabolism (Figure 4D). Besides, there were significant differences in the F/B ratio between three enterotypes (*p* = 0.0076, Kruskal–Wallis test), ratio of F/B in enteroype1 and enteroype2 were significantly higher than that in enterotype3 (Figure 4E), further revealing that anthocyanins could change energy metabolism of the gut microbiome, which may account for the changes in the structure and diversity of the gut microbiome induced by oral supplement with anthocyanins.

### 3.6. Oral Supplement with Anthocyanin and Anthocyanin Combo Changes the Production of Fecal SCFAs

Microbial fermentation of nondigestible dietary carbohydrates (NDCs) mainly leads to the production of SCFAs, among which propionate and butyrate in particular exert a wide range of health-promoting functions [35]. Thus, we also collected mice feces for SCFAs assay. As illustrated in Figure 5A, the total content of SCFAs was significantly reduced in the α-PD-L1 group in contrast with the control group (*p* = 0.0474, unpaired *t*-test with Welch’s correction), which was partially recovered when PD-L1 blockade was combined with LCP–chitosan or bilberry anthocyanins. Furthermore, oral supplement with bilberry anthocyanins increased both the concentration and the proportion of butyrate (Figure 5C–H).

### 3.7. Bilberry Anthocyanins Enhance Tumor Immune Infiltration

Finally, the percentage of CD4^+^ or CD8^+^ T cells in tumor tissues was analyzed by flow cytometry. As detailed in Figure 5I, the proportion of lymphocytes to total cells was increased in all α-PD-L1 treatment groups, indicating enhanced tumor immune infiltration. Besides, there were smaller intragroup differences in the proportion of lymphocytes to total cells in the α-PD-L1/anthocyanin group and the α-PD-L1/anthocyanin combo group, which was consistent with the observation that these two groups had a smaller intragroup difference in tumor volume (Appendix A). Meanwhile, compared with the PD-L1 blockade alone, treatment with PD-L1 blockade combined with anthocyanins or anthocyanin combo showed a higher proportion of CD8^+^ T cells to total lymphocytes from tumor tissues (Figure 5K), which was associated with better control of tumor growth.

## 4. Discussion

The discovery that gut microbiome influences the clinical outcomes of ICIs has attracted immense efforts on manipulating the gut microbiome to enhance therapeutic response [10]. On the other hand, great efforts are needed to explore the interaction between microbiome and ICIs. For example, there are reports that the obese patients had a better outcome with ICIs [36,37,38,39]. Different gut microbiome signatures were observed between obese people and lean people [40] and the gut microbiome had been involved in the onset of obesity-related disorders [41]. Dietary anthocyanins have been found a role against obesity and inflammation [42] and in metabolic changes [43]. Therefore, the elucidation of the role of anthocyanins in enhancing ICIs efficiency, as observed in the present study, should be put into the context of both the metabolism and immune system. In this study, we found that oral supplement with anthocyanins or anthocyanin combo enriched the subdominant bacteria, and the PD-L1/anthocyanin combo group achieved the highest efficiency in the α-PD-L1 treatments. The addition of anthocyanins in the α-PD-L1 treatment showed an overrepresentation of *Lachnospiraceae* and *Ruminococcaceae*, and a higher F/B ratio. *Lachnospiraceae* and *Ruminococcaceae* belong to the order *Clostridiales* within the class *Clostridia* of the phylum *Firmicutes*, and many members of this family are butyrate-producing species [35,44]. These structure changes in the gut microbiome contributed to the increased production of SCFAs in anthocyanins treatment groups, especially butyrate. Butyrate is mainly metabolized by colonic cells, and it exerts anticarcinogenic and anti-inflammatory effects, and can improve intestinal mucosal barrier function [45]. Therefore, SCFAs produced by *Lachnospiraceae* and *Ruminococcaceae* may play a critical role in enhancing the α-PD-L1 mAb efficiency by oral supplement with bilberry anthocyanins.

It is worthy to note that routinely used agents may interfere with gut microbiome and thus the response to ICIs. For example, proton pump inhibitors (PPIs) used for potent gastric acid suppression have been found to perturb the gut microbiome, leading to dysbiosis and decreased efficiency of treatment with ICIs [46,47,48]. More recently, Kim et al. reported that antibiotics are associated with poor progression-free survival (PFS) and overall survival (OS) in patients with solid cancers treated with ICIs [49]. The present study shown that LCP–chitosan significantly increased the relative abundance of the *Muribaculaceae*, one of the most abundant families in the gut microbiome, while decreased the species diversity. We note that the efficiency of PD-L1 blockade treatment was not obviously influenced in comparison with the α-PD-L1 group, indicating that there are other gut microbiome signatures contributing to the efficiency of PD-L1 blockade treatment, in addition to the species diversity. The application of anthocyanin combo restored the distorted gut microbiome caused by LCP–chitosan, and, in the meantime, achieved the highest therapeutic efficiency of ICIs. The elucidation of the further improvement of ICIs efficiency by the anthocyanin combo, as compared to the anthocyanins treatment that also yielded an increased concentration of butyrate and enhanced tumor immune infiltration, needs additional investigation. Collectively, these findings demonstrate again the connection between gut microbiome with the therapeutic efficiency of ICIs and, more importantly, indicate unprecedented opportunities to manipulate gut microbiome favorable for ICIs with natural nutrients, e.g., probiotics, as stimulants.

## 5. Conclusions

In the present study, we proposed an idea in terms of biostimulation of the gut microbiome, that is, using natural nutrients to manipulate the gut microbiome so to enhance the therapeutic efficiency of PD-L1 blockade. We demonstrated this idea using anthocyanins and anthocyanin combo containing chitosan and LCP, respectively. It was shown that the oral administration of bilberry anthocyanins showed an overrepresentation of *Lachnospiraceae* and *Ruminococcaceae* in which many members are butyrate-producing species, which was consistent with the increased concentration and proportion of butyrate. The bilberry anthocyanins could change the energy metabolism of the gut microbiome, which may account for the regulation of the gut microbiome. Oral supplementation with bilberry anthocyanins during the PD-L1 blockade treatment showed enhanced tumor immune infiltration, which was associated with better control of tumor growth. The stimulating effects of bilberry anthocyanins were further intensified via using an anthocyanin combo. Although the underpinning mechanisms of enhanced tumor immune infiltration and the antitumor efficiency of the PD-L1 blockade by anthocyanin and anthocyanin combo remain to be discovered, our results reconfirmed the connection of gut microbiome with ICIs and revealed the opportunities of biostimulation of gut microbiome in checkpoint inhibitor therapy.

## Figures and Tables

**Figure 1 microorganisms-08-00175-f001:**
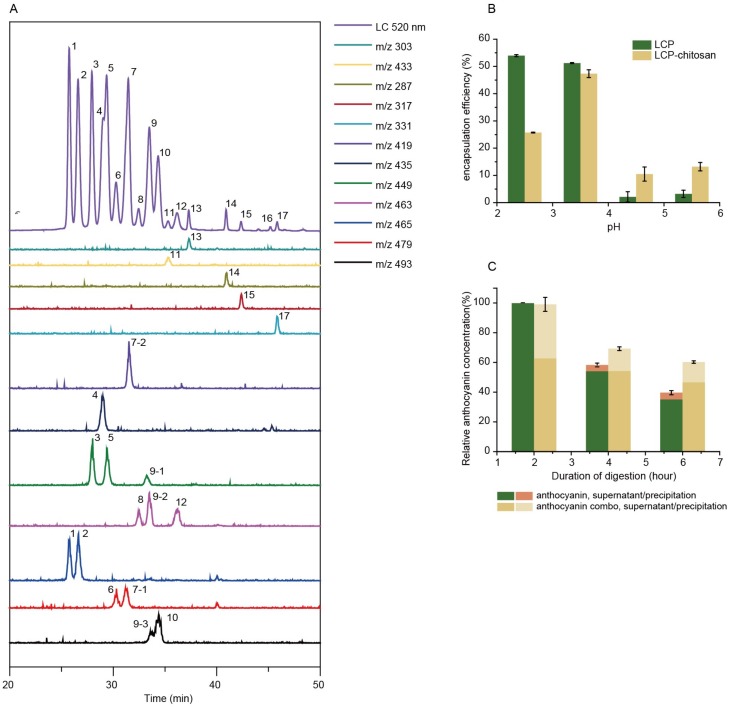
Main components and digest stability. (**A**) High-performance liquid chromatography–ultraviolet-visible spectroscopy-tandem mass spectrometry (HPLC-UV/vis-MS/MS) chromatograms (detected at 520 nm) of bilberry extracts. (**B**) Chitosan and low molecular citrus pectin (LCP) fabricated polyelectrolyte complexes at different pHs. (**C**) Relative content of anthocyanins of bilberry extracts and anthocyanin combo (anthocyanin–LCP–chitosan) in simulated gastric fluid (SGF) with pepsin at pH 2.0 for 2 h (**A**) and subsequently in simulated intestinal fluid (SIF) with pancreatin at pH 6.8 for 4 h.

**Figure 2 microorganisms-08-00175-f002:**
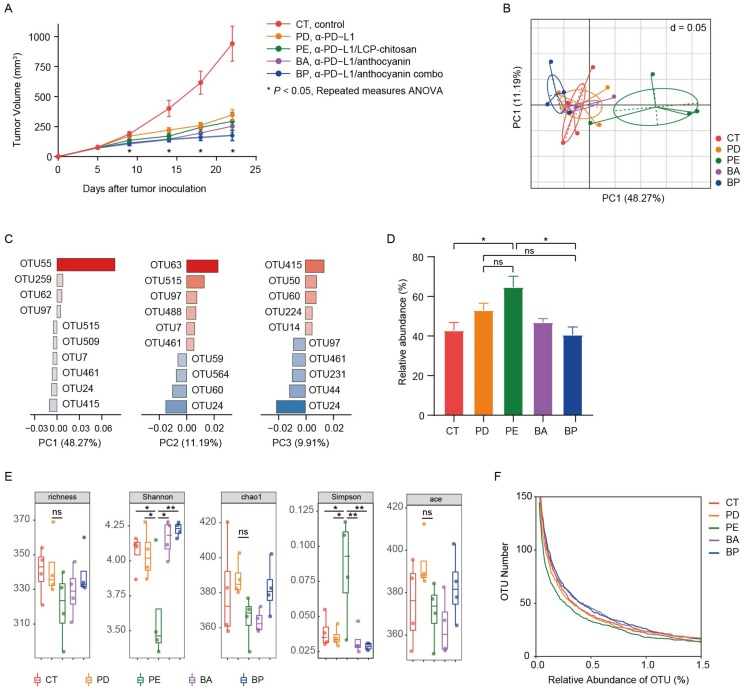
Oral administration of bilberry anthocyanins enhances anti-PD-L1 efficiency and changes the mouse gut microbiome structure. (**A**) C57BL/6 mice were inoculated subcutaneously with 5 × 10^5^ MC38-OVA cells and treated with 200 μg of the immunoglobulin G2b (IgG2b) isotype or the α-PD-L1 mAb on days 7 and 10. CT, the IgG2b isotype control; PD, the α-PD-L1 group; PE, the α-PD-L1/LCP–chitosan group; BA, the α-PD-L1/anthocyanin group; BP, the α-PD-L1/anthocyanin combo group. Repeated measures ANOVA (time × tumor volume) and Sidak’s multiple comparisons test were used to test mouse tumor growth between groups. Data indicates mean ± SEM, *n* = 6 per group, ** p* < 0.05. (**B**) Beta-diversity analysis of microbial communities by using principal component analysis (PCA) based on OTUs. (**C**) The associated contribution plot illustrating the key OTUs that impact the PCA clustering. (**D**) Significant increase of *Muribaculaceae* in the α-PD-L1/LCP–chitosan group. (**E**) For alpha diversity, richness, chao1, Shannon, Simpson, and ace indices were calculated using random subsamples of 35,043 sequences per sample, Data indicates mean ± SEM. * *p* < 0.05, ** *p* < 0.01 (Student’s *t*-test). (**F**) OTU richness under different thresholds of relative abundance.

**Figure 3 microorganisms-08-00175-f003:**
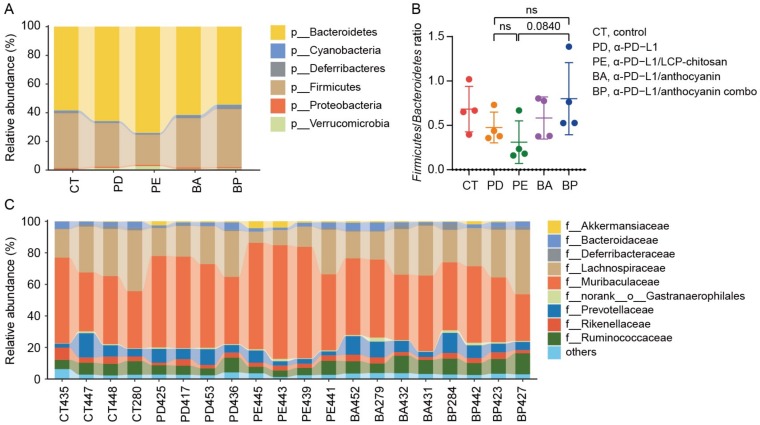
Composition of the gut microbiome at different taxonomic levels. (**A**) Phylum level taxonomic composition of bacterial communities in each group. (**B**) The *Firmicutes* to *Bacteroidetes* ratio. (**C**) Taxonomic composition of the bacterial communities in each group at family level.

**Figure 4 microorganisms-08-00175-f004:**
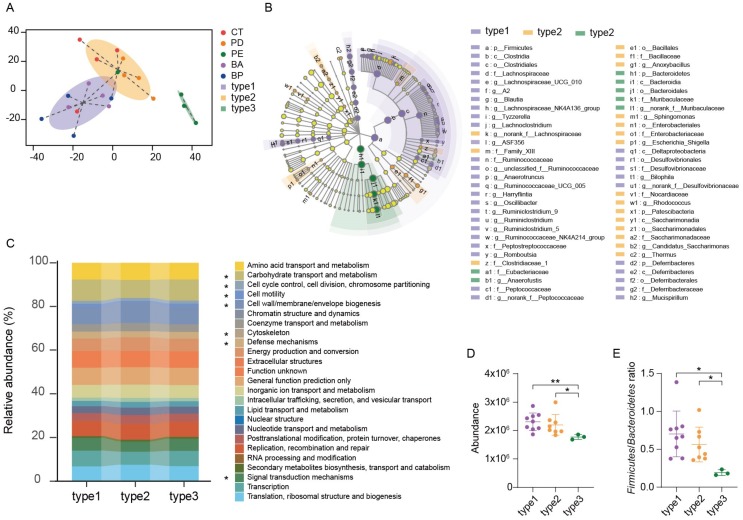
Enterotype analysis and functional differences between different enterotypes. (**A**) Gut microbiome was clustered into three enterotypes at OTU-level using Jensen–Shannon distance, and the graphs for enterotype clusters indicate the first two principal coordinates. (**B**) Linear discriminant analysis effect size (LEfSe) was used to differentiate the three enterotypes. (**C**) PICRUSt prediction of functional profiling of the microbial communities from these three enterotypes based on the 16SrRNA gene sequences. (**D**) Difference in the function of carbohydrate transport and metabolism and (**E**) The *Firmicutes* to *Bacteroidetes* ratio were tested using Student’s *t*-test, * *p* < 0.05, ** *p* < 0.01.

**Figure 5 microorganisms-08-00175-f005:**
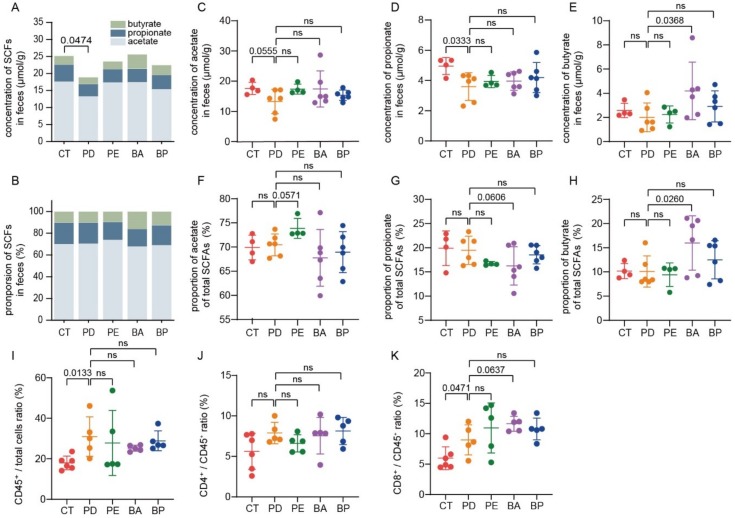
Bilberry anthocyanins promote the production of short-chain fatty acids (SCFAs) and tumor immune infiltration. The total content of SCFAs (**A**) and the proportions of individual to total SCFAs in feces. (**B**) Quantification of the content of acetate (**C**), propionate (**D**) and butyrate (**E**) in feces by gas chromatography. The proportions of acetate (**F**), propionate (**G**) and butyrate (**H**) to total SCFAs in feces. Quantification of total T lymphocytes (**I**), CD4^+^ (**J**) and CD8^+^ (**K**) T cells in tumor tissues by flow cytometry. Differences were tested using Student’s *t*-test; ns, not significant.

**Table 1 microorganisms-08-00175-t001:** Peak identification of anthocyanins in Mirtoselect^®^ bilberry extract using HPLC-UV/vis-MS/MS.

Peak #	Species	Formula	Molecular Ions (m/z)	Fragment Ions MS^2^ (m/z)
1	Delphinidin-3-O-galactoside	C_21_H_21_O_12_	465	303
2	Delphinidin-3-O-glucoside	C_21_H_21_O_12_	465	303
3	Cyanidin-3-O-galactoside	C_21_H_21_O_11_	449	287
4	Delphinidin-3-O-arabinoside	C_20_H_19_O_11_	435	303
5	Cyanidin-3-O-glucoside	C_21_H_21_O_11_	449	287
6	Petunidin-3-O-galactoside	C_22_H_23_O_12_	479	317
7	Petunidin-3-O-glucoside/Cyanidin-3-O-arabinoside	C_22_H_23_O_12_/C_20_H_19_O_10_	479/419	317/287
8	Peonidin-3-O-galactoside	C_22_H_23_O_11_	463	301
9	Petunidin-3-O-arabinoside/Peonidin-3-O-glucoside/Malvidin-3-O-galactoside	C_21_H_21_O_11_/C_22_H_23_O_11_/C_23_H_25_O_12_	449/463/493	317/301/331
10	Malvidin-3-O-glucoside	C_23_H_25_O_12_	493	331
11	Peonidin-3-arabinoside	C_21_H_21_O_10_	433	301
12	Malvidin-3-O-arabinoside	C_22_H_23_O_11_	463	331
13	delphinidin	C_15_H_11_O_7_	303	303, 229
14	Cyanidin	C_15_H_11_O_6_	287	287, 137
15	Petunidin	C_16_H_13_O_7_	317	203, 245
16	Peonidin	C_16_H_13_O_6_	301	201, 229
17	Malvidin	C_17_H_15_O_7_	331	315, 242, 287

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
