# Peer review of "Biostimulating Gut Microbiome with Bilberry Anthocyanin Combo to Enhance Anti-PD-L1 Efficiency against Murine Colon Cancer"

_microorganisms, 2020, doi:10.3390/microorganisms8020175_

Round 1
Reviewer 1 Report
The present manuscript reports on biostimulating gut microbiome with bilberry anthocyanin combo to enhance anti-PD-L1 efficiency against murine colon cancer.
This is a very hot topic in immuno-oncology and growing literature suggest a crucial role for gut microbiome on the efficacy of PD(L)-1 inhibitors in cancer patients. Therefore, strategies that stimulates gut microbiome or favorably alter its composition (as for example with fecal transplant) are under active evaluation.
I suggest the acceptance of the manuscript with major revision.
Some comments:
Recently, some studies have suggested the obese patients have a better outcome with ICIs compared with other patients (Cortellini A, et al. J Immunother Cancer. 2019; Xu H, et al. Int Immunopharmacol. 2019; Ichihara E, et al. Lung Cancer 2020; Kichenadasse G, et al. JAMA Oncol 2020). Interestingly, gut microbiota has been shown to be a key contributor involved in the onset of obesity-related disorders and in a recent paper (Depommier C, et al. Nat Med 2019) reported that supplementation with Akkermansia muciniphila improves several metabolic parameters. As reported in your paper, the presence of the anaerobic commensal Akkermansia Muciniphila is more common in responders to ICIs.Interestingly, dietary anthocyanins seem to have a role against obesity and inflammation [Lee YM, et al. Nutrients. 2017] and in metabolic changes [Overall J, et al. Int J Mol Sci. 2017].
Please include a comment on this emerging interesting topic.
In cancer patients, gut microbiome can be influenced by several different agents that are commonly used, as for example proton pump inhibitors (PPIs). Potent gastric acid suppression using PPIs has important effects on human health and may perturb microbial communes, leading to dysbiosis and an increased risk of enteric infection and diarrhea in humans. Recently, some studies have evaluated the impact of PPI use on immune checkpoint inhibitors efficacy [Homicsko et al. ESMO Immuno-Oncology 2018; Trabolsi A, et al. ASCO annual meeting 2019; Mukherjee S, et al. J Oncol Pharm Pract. 2019]. A comment on the results of these studies can be useful
Reviewer 2 Report
In this study, authors are examining the effect of biostimulating of the gut microbiome and its effect on anti-PDL1 efficacy. This study is of significant interest to the scientific community but it requires major revisions before it would be acceptable for publication is Microorganisms.
1- Use the same colour code across the paper (in figures). At its current format data is confusing to the readers. Follow the same order for all the treatment groups as the tumor growth inhibition data.
2- You need to compare all the groups to the anti-PDL1 alone as your proper control across the paper. Authors use different groups as controls that is not acceptable.
3- The diversity data (figures 2d and e) needs to be compared to anti-PDL1 group. The current conclusions are not correct.
4- The decrease in diversity in the PE group is not related to reduced anti-PDL1 efficacy. Therefore, the conclusions are not valid. Simply, diversity does not play any role here in improving the efficacy of anti-PDL1. This needs to be changed in the manuscript and should be reflected in the conclusions.
5- Figure 3b, required statistical analysis.
6- Figure 5 like other figures, the proper control is the anti-PDL1 alone. Re do the analysis and write the conclusions based on that.
7- I am not certain whether the conclusions that the authors are making would hold up when data is reanalyzed and compared to anti-PDL1 alone.
8- There are typos in the manuscript.
Round 2
Reviewer 1 Report
The authors addressed all the issues I reported in the first manuscript draft. I suggest the acceptance of the manuscript without any further revision
Reviewer 2 Report
As I predicted most of the groups when compared to the anti-PDL1 group did not show any significant difference. Authors have modified their statements regarding their findings. This is now acceptable for publication.